# What Can the Gut Microbiota of Animals Teach Us about the Relationship between Nutrition and Burden of Lifestyle Diseases?

**DOI:** 10.3390/nu16111789

**Published:** 2024-06-06

**Authors:** Denise Mafra, Natália A. Borges, Beatriz G. Baptista, Layla F. Martins, Gillian Borland, Paul G. Shiels, Peter Stenvinkel

**Affiliations:** 1Graduate Program in Medical Sciences and Graduate Program in Nutrition Sciences, Federal Fluminense University (UFF), Niterói 24020-141, Brazil; biabia.baptista01@gmail.com; 2Graduate Program in Biological Sciences—Physiology, Federal University of Rio de Janeiro (UFRJ), Rio de Janeiro 21941-909, Brazil; 3Graduate Program in Food, Nutrition, and Health, Institute of Nutrition, State University of Rio de Janeiro (UERJ), Rio de Janeiro 21941-909, Brazil; nat_borges_@hotmail.com; 4Department of Biochemistry, Institute of Chemistry, University of São Paulo (USP), São Paulo 05508-220, Brazil; layla@iq.usp.br; 5School of Molecular Biosciences, University of Glasgow, Glasgow G12 8QQ, UK; gillian.borland@glasgow.ac.uk (G.B.); paul.shiels@glasgow.ac.uk (P.G.S.); 6Division of Renal Medicine, Department of Clinical Science, Technology and Intervention, Karolinska Institutet, 17165 Stockholm, Sweden; peter.stenvinkel@ki.se

**Keywords:** biomimetics, food, chronic diseases, gut microbiota

## Abstract

The gut microbiota performs several crucial roles in a holobiont with its host, including immune regulation, nutrient absorption, synthesis, and defense against external pathogens, significantly influencing host physiology. Disruption of the gut microbiota has been linked to various chronic conditions, including cardiovascular, kidney, liver, respiratory, and intestinal diseases. Studying how animals adapt their gut microbiota across their life course at different life stages and under the dynamics of extreme environmental conditions can provide valuable insights from the natural world into how the microbiota modulates host biology, with a view to translating these into treatments or preventative measures for human diseases. By modulating the gut microbiota, opportunities to address many complications associated with chronic diseases appear. Such a biomimetic approach holds promise for exploring new strategies in healthcare and disease management.

## 1. Introduction

“All disease begins in the gut”—(Hippocrates of Kos).

Although the analysis of gut microbial composition in humans is still limited, a range of studies has demonstrated that modulating the composition of the gut microbiota alters metabolism. Indeed, the microbiota has a crucial role in health and development, e.g., through the production of vitamins and short-chain fatty acids (SCFA), modulation of the immune system, and protection against pathogenic bacterial species colonizing the gut. The composition of the gut microbiota changes according to intrinsic and extrinsic exposome factors, including diet (the primary modulator), genetics, drugs, stress, geographical area, aging, etc.), and alterations in this composition can be related to health status [1,2].

In humans, the diseasome of aging (i.e., age-related non-communicable diseases underpinned by a common component of dysregulated aging), which includes diabetes, cardiovascular disease, cancer, autoimmune disorders, neuropsychiatric disorders, and chronic kidney disease, is associated with gut dysbiosis [1]. Correspondingly, microbial dysbiosis has been ascribed as a new hallmark of aging. Notably, modulation of the effects of the microbiome using food may open up new therapeutic and interventional avenues for treating chronic disease conditions [3].

Biomimetics, rooted in seeking innovation from nature, is a concept integral to scientific exploration [4]. Albert Einstein’s quote “Look deep into nature, and then you will understand everything better” underscores nature’s profound insights, highlighting how studying the natural world can deepen our comprehension of life’s intricacies. Biomimetic studies illuminate principles and strategies found in nature that have evolved through natural selection, offering solutions to contemporary challenges. Researchers investigate how evolution, driven by natural selection, has equipped animals and plants with adaptations to thrive in their environments. They explore nature’s wisdom for solutions to environmental sustainability amidst human impact [5,6]. Understanding how animals adapt to their nutritional environments can provide valuable insights for developing strategies to mitigate gut dysbiosis by targeting the foodome. This approach is especially relevant when considering the interaction with the epigenetic landscape, a dynamic feature responsive to real-time environmental changes that can be regulated by bioactive compounds derived from food [1,3,7].

Biomimetic research addresses key questions around the interplay between the foodome and the microbiota and health. How, for example, do these features interact to sustain the mechanisms underpinning good health? Understanding the role of the gut microbiota in ruminants’ production of short-chain fatty acids (SCFA) from fiber fermentation to ameliorate inflammatory processes and expression of the cytoprotective transcription factor nuclear factor erythroid 2-related factor 2 (Nrf2) to promote cellular resilience to stress can lend support to novel treatment strategies [8]. Additionally, the resilience of bears during hibernation, elephants’ cancer resistance, and the giant salamander’s resilience to global warming offer valuable insights into the regulation of the cellular processes that are dysregulated in the “diseasome of aging”. This narrative review, adopting a biomimetic perspective, will explore the prospect of modulating the microbiota to alleviate specific phenotypes associated with chronic disease.

## 2. The Emergence of an Industrialized Microbiota

The gut microbiota of mammals is remarkably diverse, comprising bacteria, fungi, protozoa, archaea, yeasts, and bacteriophages. The human gut microbiota has co-evolved with our species to create a holobiont, whereby the microbial community has a composition honed over millions of years of evolution by natural selection. However, the rapid industrialization and urbanization (air and water pollution, heat stress, and microplastic exposure) of human society during the Anthropocene has dramatically altered the composition and diversity of the gut microbiome at a rate that has outpaced natural symbiotic selective processes. This has resulted in a wave of “burden of lifestyle” diseases driven by a sedentary lifestyle and industrialized environmental changes, inclusive of a Western-style ultra-processed diet. In addition, another issue of this era is the pollution of drinking water with xenobiotic substances, such as dyes, pharmaceuticals, personal care products, endocrine-disrupting compounds, pesticides, and polycyclic aromatic hydrocarbons. Drinking water is among the items consumed in the largest amount. It may be considered an important factor in shaping the gut microbiome, affecting the composition, gene expression, and function of the gut bacteria, consequently impacting the host health [9,10]. All these are associated with various degrees of microbial dysbiosis. The adverse health consequences of an industrialized microbiota are not yet fully realized [1,11,12].

Comparative studies of gut microbiomes across non-Westernized and industrialized populations have revealed stark differences. The traditional human microbiome, present in hunter–gatherer and subsistence farming communities, is characterized by high microbial biodiversity and ancient microbial lineages that are increasingly rare in modern industrialized societies [13,14]. In contrast, the industrialized human microbiome is marked by reduced richness and a relative predominance of microbes adapted to the Western diet and lifestyle [12].

This microbiome depletion and homogenization can be attributed to several key factors associated with industrialization. The shift towards processed nutrient-poor foods high in fat, sugar, and synthetic additives has profoundly reshaped the gut microbial community, favoring microbes adept at metabolizing the components of the industrialized diet over fiber-fermenting plant polysaccharide-degrading bacteria [15]. For example, additives from ultra-processed food can impair the gut microbiota composition and lead to alterations in the gut microbiota–immune system axis [16].

Improved sanitation, ubiquitous use of antimicrobials, and reduced exposure to diverse environmental microbiomes have also diminished microbial acquisition and transmission, especially in early life stages critical for optimal microbiome development [17]. Additionally, diminished physical activity and time spent in natural environments decrease contact with environmentally derived microbes that can seed the human gut [18]. Epidemiological studies have linked the industrialized microbiome profile to higher risks of immune-mediated, metabolic, and neurological disorders. The loss of microbial diversity and ancestral microbial species compromises the microbiome’s protective, metabolic, and signaling functions that maintain human health [12]. Notably, the depletion of SCFA-producing bacteria is associated with inflammation, insulin resistance, and cognitive impairment [19,20].

Therefore, restoring a biodiverse ecologically stable microbiome is a crucial public health priority. Potential strategies include promoting traditional dietary practices, increasing exposure to green spaces and environmental microbiomes, and developing targeted probiotic and prebiotic interventions [12,18]. Ultimately, recovering a healthy ancestral-like gut ecosystem may be essential for reclaiming our evolutionary-tuned physiology and preventing the rising tide of modern non-communicable diseases.

## 3. What Does the Natural World Tell Us about an Industrialized Microbiota?

By examining the microbiomes of humans and other species in natural non-industrialized environments, we can glean valuable insights into the consequences of living in the Anthropocene on the human microbiome. The natural world provides a window into the types of microbial communities that our species co-evolved with, offering critical context for understanding the dysbiosis observed in industrialized societies. This is a powerful comparative lens to understand the industrialized human microbiome as a radical human-driven perturbation in an ancient symbiosis [21].

Comparative studies of hunter–gatherer and traditional agricultural communities have consistently demonstrated remarkable microbial diversity and the presence of ancient microbial lineages that are increasingly rare in Westernized populations [13,22]. For example, the Hadza of Tanzania, one of the last remaining hunter–gatherer groups, harbor a gut microbiome rich in bacteria like Treponema and Prevotella, which are virtually absent in urban industrialized settings [13]. A study examining 54 mammalian species revealed that the vulnerable giant anteater exhibited the highest richness in gut microbiota, while carnivores displayed the lowest. This finding suggests that a diverse microbiota may not be necessary for digesting meat-rich diets compared with herbivorous diets. Moreover, animals with foregut physiology demonstrated higher richness and greater microbiota diversity, indicating the complexity of gut physiology [23]. The rumen, housing microorganisms responsible for fermenting fibers and potentially altering dietary toxins, such as tannins [24], has a crucial role in herbivores. Early studies involving germ-free sheep revealed that these animals could not survive long after transitioning from milk to solid food, highlighting the vital role of gut microbiota in herbivorous species [25]. Similarly, analysis of wild great ape microbiomes revealed a level of diversity far exceeding that of human microbiomes, even in non-Westernized human populations [26]. These ancestral microbial communities play crucial roles in regulating host health, particularly immunity. In captivity, primate microbiomes rapidly “humanize” and lose significant diversity [27].

The natural world also provides insights into the specific mechanisms by which industrialization disrupts the human microbiome. Studies of wild mammals have shown that urbanization, pollution, and agricultural practices can directly perturb environmental microbiomes [28,29]. Microbes essential for healthy ecosystem functioning are displaced by opportunistic pathobiont-like species better equipped to thrive in contaminated disturbed habitats. Importantly, experimental animal studies have demonstrated that exposing hosts to these industrialized environmental microbiomes can recapitulate the dysbiotic disease-associated gut microbial profiles observed in humans [12,29]. This highlights how environmental microbiome deterioration due to human activities may be a driver of microbiome depletion and homogenization in industrialized populations.

Additionally, wild primate studies reveal the critical role of dietary fiber and plant secondary metabolites in shaping healthy gut microbial communities [30]. The dramatic shift towards processed fiber-poor foods in industrialized human diets likely removes these essential microbial substrates, contributing to the proliferation of microbes maladapted to a plant-rich diet. Reconnecting with environmental microbiomes and dietary patterns that align with our evolutionary heritage may be essential for restoring microbial diversity and the holistic health benefits it confers [11].

## 4. Insights Gleaned from Studying the Gut Microbiota of Specific Animals

Nature is full of interesting natural animal models developed during evolution that provide excellent models for studies about the impact of nutritional habits on the microbiome, evolutionary adaptation, and metabolic functions [1]. Table 1 outlines some distinctive features of certain animals concerning gut microbiota metabolism. This microbiota has evolved as a surrogate organ, connecting human immune function, diseases, and dietary habits [31]. *Salutogenic bacteria* such as *Bacteroidaceae* and *Bifidobacteriaceae* have remained human commensals across ages [26]. In this review, we will focus on some species (naked mole rats (NMRs), pandas, elephants, tigers, and koalas) in which alterations in the gut microbiota have proven to play a role in the adaptive development of a protective phenotype (Figure 1).

Naked mole rats (NMRs) are subterranean long-living rodents that exhibit coprophagic habits, both autocoprophagy and allochoprophagy. This intriguing behavior is believed to play a role in maintaining their health, nutrient balance, and eusocial behaviors. In the underground nest, there may be changes in the composition of fecal bacteria due to oxygen deprivation and the mixing of bacteria from the feces and surrounding microorganisms. Coprophagy may be an ancient mechanism for acquiring microorganisms and preventing uniformity in their bacterial microbiome. When coprophagia occurs, animals may also re-ingest the vitamin K2 initially produced by these gut bacteria. Vitamin K2 is important for various bodily functions, including blood clotting and bone health. On the other hand, animals engaging in coprophagic habits may be more vulnerable to microbial exposure [50]. In contrast, NMRs demonstrate a reduced bacterial load and diminished inflammatory potential in their gut microbiota compared with mice and humans. This distinction is likely attributed to the constant expression of CD14 on their granulocytes and the presence of the cathelicidin gene, which exhibits potent antimicrobial properties [51,52,53,54]. Understanding the microbiome and immune system characteristics can broaden our knowledge of the evolutionary characteristics of animals that serve as models for longevity and cancer prevention [50,55].

It is noteworthy that certain unique compositions of gut microbiota found in NMR, including the elevated diversity of *Spirochaetaceae* and *Mogibacteriaceae*, resemble those observed in human models of a healthy gut microbiome, such as centenarians and Hadza hunter–gatherers. Three days of a Hazda hunter foraging diet causes a massive increase in microbiome diversity [56]. Thus, we could make the effort to improve our gut health by re-wilding our diet and lifestyle. NMRs’ unique metabolism involves utilizing soil sulfate to accept terminal electrons, sustaining an anaerobic oxidative gut metabolism [33]. This sulfate-dependent metabolism suggests a strategic role for sulfhydration in protecting against disease and premature aging. Furthermore, it highlights the importance of functional processes in host–microbiota interactions alongside microbial composition and abundance [57].

The metabolome sheds light on microbiome–host interactions that contribute to the longevity of NMRs. Compared with mice, they exhibit lower levels of 3-Indole propionic acid, likely of bacterial origin, suggesting potential variations in the microbiota that may explain some of the metabolic differences between the two species [58]. Additionally, the gut microbiota of NMR is characterized by a significant abundance of Bacillus megaterium, known for its efficiency in polyamine biosynthesis [51,59]. Polyamines, such as spermidine, are recognized for their ability to induce autophagy, a cellular process linked to maintaining good health and promoting longevity. Multiple studies have reported enhanced autophagy in NMR [58,60,61].

Another remarkable characteristic of NMRs is their extraordinary tolerance to hypoxia. They can survive up to 18 min of complete oxygen deprivation thanks to metabolic adaptations that involve rewiring glycolysis [62]. The connections between microbiota and hypoxia are fascinating, involving a dynamic exchange of gaseous signaling mediators produced by bacterial and intestinal metabolisms [63]. In the colon, where the partial pressure of oxygen (pO2) ranges from 0.4 to 2.0% (compared with 21% at sea level and 10% in tissues), intestinal hypoxia fosters the growth of anaerobic bacteria that produce gases like NO, CO, H_2_S, and CO_2_, known to activate hypoxia-inducible factors (HIFs). HIFs regulate genes involved in various adaptations to hypoxia, resulting in increased mucin production, β-defensins from mucus, and the regulation of tight junction proteins such as claudin-1. However, increased activation of HIF may be associated with diseases like inflammatory bowel disease and colorectal cancer [63]. A recent study has demonstrated that C57BL/6 J mice exposed to hypoxic conditions (simulated high-altitude hypoxia) exhibited alterations in their gut microbiota composition (increasing Akkermansia and reducing Firmicutes-to-Bacteroidetes ratio and Bifidobacterium) compared with the normoxic group, showing a beneficial response to environmental stress [64]. This is also seen in blue sheep [65], European mouflon, and Tibetan antelope [66].

Although evolutionary evidence suggests that ancestral pandas were carnivorous, they adapted to a herbivorous diet over time. Pandas, with a digestive system typical of carnivores, possess a gut microbiota seemingly ill-prepared for fiber fermentation despite their almost exclusive bamboo diet [36,67]. However, they exhibit an increased abundance of the Proteobacteria phylum, capable of lignin degradation—the primary bamboo component—potentially serving as a compensatory mechanism. Streptococcus alactolyticus is the predominant bacterial species in giant pandas’ gut microbiota. This bacterium plays a crucial role in essential amino acid biosynthesis within the gut microbiota, indicating its involvement in the host’s adaptation to a bamboo-centric diet [36]. In tandem with the compensatory role played by Proteobacteria in pandas, where the diet (bamboo) contrasts with the gut microbiota typical of carnivores, Bifidobacterium, known for lactose fermentation, could have aided in the human adaptation to milk consumption. Remarkably, alleles in the gene linked to lactase persistence in humans exhibit a notable correlation with the gut microbiota composition [25,68]. Research has revealed shifts in the metabolites, composition, and function of the gut microbiome as giant pandas transition from a milk-rich diet in cubs to a bamboo-focused diet in the early and adult stages. Furthermore, a microbiome’s involvement in age-related metabolism in giant pandas has been proposed. Notably, substantial variations in bile acid content were observed among different age groups, including cubs, young, adult, and old, with a notable increase in bile acid metabolites detected in the fecal samples of older pandas. This increase could be linked to the higher incidence of lipid metabolism disorders observed in these animals. Furthermore, there was a positive correlation between bile acids glymodeoxycholic, taurodeoxycholic, and taurodeoxycholic and Lactobacillus and Bifidobacterium species [69]. These bacteria can produce enzymes that facilitate the conversion of primary to secondary bile acids, a process linked to compromised intestinal barrier function and increased production of reactive oxygen species [70,71]. This finding aligns with the oxidative stress and inflammation metabolome profile observed in older giant pandas [69].

The gut microbiome of Asian elephants exhibits a high proportion of Firmicutes, known for their lignocellulose-degrading enzymes, similar to the digestive profile observed in giant pandas [72]. Moreover, the gut microbiota of elephant calves is primarily composed of milk-fermenting taxa, such as the Bifidobacterium genus of the Bifidobacteriaceae family and Akkermansia genus. As elephants mature into subadults and adults, Firmicutes become the dominant phylum, followed by Bacteroidetes and Actinobacteria [38]. Research tracking the transition of gut microbiota profiles in Asian elephants throughout different stages of growth and development has revealed that microbiota diversity is lowest during infancy, remains stable in adulthood, and slightly decreases during the geriatric phase [38], matching what is observed in humans. The decline in diversity among geriatric elephants may stem from losing their last molar teeth, resulting in larger food particles in their feces, which could lead to less efficient food utilization by gut bacteria [38,73]. Intriguingly, it has been suggested that the decrease in human microbiome diversity may be more closely linked to age-related frailty than chronological age. Furthermore, during this stage of life, immune health also experiences instability, suggesting a symbiotic relationship between gut microbiota, the immune system, and overall health status [74].

Tigers present a higher abundance of Fusobacterium, commonly found in carnivorous predators and associated with red meat consumption and digestion. Researchers compared older and cub tigers receiving meat and milk, respectively, and they found that meat-fed tigers showed increased abundances of Fusobacterium, and the milk-fed tigers presented at the genus level a high abundance of Bifidobacterium, Escherichia/Shigella, and Lactobacillus [40]. Intriguingly, when tigers are fed milk, the abundance of Fusobacterium decreases [40]. Fusobacterium is associated with some diseases, such as acute appendicitis and colon cancer in humans [40,75]. This is interesting, considering that the high abundance of Fusobacterium is seen in meat eaters [76]. Further comparative studies may provide valuable insights into the relationship between red meat consumption and human cancer risk. As tiger cubs fed goat milk exhibit a reduced proportion of Bifidobacterium and Lactobacillus compared with those nursed by their mothers, this suggests that varying dietary patterns can alter the gut microbiota composition in these animals [77]. It is noteworthy that tiger cubs raised in captivity and requiring artificial feeding have a higher mortality rate attributed to gastrointestinal diseases and associated alterations in gut microbiota. Addressing this issue could provide important solutions for improving their health and survival in captivity [40].

The gut microbiome composition of South China tigers kept in the same zoo and fed artificially mainly consists of Fusobacteria, Firmicutes, Bacteroidota, Proteobacteria, and Actinobacteriota. The gut microbiome of these tigers undergoes significant changes as they age, progressing through developmental (cub), transitional (subadult), and stable (adult) phases. Initially, at five months of age, Fusobacteriota dominates the gut microbiome, but, as the tigers grow older, Firmicutes become the predominant phylum. The ratio of Firmicutes to Bacteroides increases between the cub and adult stages. While there are no discernible differences in the alpha diversity of the gut microbiome among juvenile, subadult, and adult South China tigers, it has been suggested that the richness and evenness of their gut microbiome may gradually increase with age [78]. Conversely, Zhu et al. [79] reported that microbial diversity and richness decrease with age in captive Amur tigers. Furthermore, there is ongoing discussion regarding the high proportion of Firmicutes and reduction in Bacteroidetes observed in obese animals compared with lean ones. Han et al. [80] observed that heavier Amur leopards, storing fat due to severe cold conditions, exhibited a higher proportion of Firmicutes than North Chinese leopards. 

The koala relies heavily on a diet composed almost entirely of foliage from eucalyptus trees, which contain various potentially harmful plant secondary metabolites. Consequently, koalas and their gut microbes have undergone co-evolution to effectively digest this low-nutrient potentially toxic diet [25]. Bacteroidetes, Firmicutes, and Cyanobacteria are co-dominant in the gut microbiota of wild koalas. However, the gut microbiota of koalas exhibits specificity according to the species of eucalyptus consumed, showing significant differences in microbial composition. Some eucalyptus species are associated with dominance of the genera Parabacteroides and/or Bacteroides, while others show dominance of an unidentified genus from the family Ruminococcaceae. Differences likely influence these variations in the gut microbiome of koalas in the protein and fiber content present in different eucalyptus diets [81]. Notably, changes in the Bacteroidetes to Firmicutes ratio in koalas may be attributed to protein and fiber content variations across different diets. Specifically, a higher prevalence of Firmicutes and a reduced relative abundance of Bacteroidetes have been associated with microbiome adaptations to increased fiber intake and decreased protein consumption during transitions from animal-based to plant-based diets [82].

Contrary to expectations, no significant overall changes are observed in the fecal microbiomes of koalas following their relocation from an area characterized by severe over-browsing and koala starvation to a mixed eucalypt forest. Despite the koalas having access to a wider variety of eucalyptus species post-relocation, their microbiomes maintain consistency with their pre-relocation composition. This suggests that the koalas can find suitable diets in the new habitat without necessitating significant microbiome adaptations. Consequently, it is proposed that the gut microbiomes of koalas are primarily shaped by their acquisition and development throughout their lives, with subsequent dietary changes exerting a comparatively minor influence [83].

## 5. Gut Microbiota in Captive versus Wild Animals

Diet plays a significant role in shaping the ancient microbial lineages and promoting the growth of specific bacteria. However, exposome factors such as ecology, life history, and physiology also contribute to the diversity and structure of the microbiota in non-human primates, even when they share the same environment [84]. Studies on various species, including tigers, elephants, giraffes, bears, monkeys, and sea lions, have shown differences in gut microbiome composition between captive and wild animals. These differences can be attributed to various factors such as diets, pharmacological interventions, increased human contact, and reduced interaction with other wild animals [85,86]. For instance, a study involving Asian elephants relocated from a semi-captive camp in Myanmar to a zoo in Japan revealed alterations in the gut microbiota profile due to anthropogenic activities, such as transportation, captivity, and deworming [86].

A comprehensive study examining the microbiota of 41 species of captive and wild mammals found that alpha diversity was generally reduced in captive mammalian families. Furthermore, captive animals displayed altered gut bacterial communities compared with their wild counterparts, except for impalas, giraffes, and antelopes. Taxonomic class analysis revealed decreased Prevotella (Bacteroidetes) and Clostridia (Firmicutes) in captive animals. Conversely, there was an increase in the relative abundance of anaerobic Bacilli such as Streptococcus luteciae and Clostridium (Firmicutes), as well as Gammaproteobacteria (Proteobacteria), which was associated with a diminished gut health status in captive animals [85]. Indeed, as diet is the primary driver of the gut microbiota, anthropogenic activities can change the gut microbiota in captive animals [87].

## 6. Gut Microbiota Changes Due to Changes in the Environment

It has been suggested that mammals have evolved mechanisms to utilize their gut microbiota as sensors, triggering adaptive responses to changes in their external environment [25]. Seasonal variations have been observed to alter the gut microbiota profiles in animals, as demonstrated in monkeys [30,42]. Similarly, altitude influences the microbiota, resulting in profiles capable of metabolizing high-fiber foods and associated with energy production from microbial metabolites such as SCFA and methane in sheep [44] and plateau pikas [45]. 

In the animal kingdom, body temperature varies significantly, ranging from 5 °C in hibernating animals to 29 °C in platypuses, with an average of 37 °C in humans [88]. Interestingly, bats increase their temperature above 40 °C during flights, and reduce it dramatically during daily torpor to around 10 °C, which can be a way to keep energy and limit the viral load since they are a natural reservoir for viruses that are not harmful to them [89]. Environmental temperatures affect microbiota composition. Some bacteria, like Escherichia coli and Enteropathogenic Yersinia, can survive in warm and cold conditions. Proteobacteria, a predominant phylum in some animals, demonstrate flexibility in response to temperature changes, with a positive association with higher temperatures. Conversely, Firmicutes correlate negatively with elevated temperatures [88,90]. The metabolic activity of microbiota in non-hibernating small mammals is believed to contribute to thermogenesis by releasing norepinephrine from the sympathetic nervous system in response to cold stimuli. This leads to increased SCFA and secondary bile acid production, stimulating G protein-coupled receptors and enhancing brown adipose tissue activity, thus promoting thermogenesis [91]. Unsurprisingly, gut microbiota plays a crucial role in the seasonal metabolic changes observed in bears, affecting glucose and fat metabolism. These findings hold promise for potential treatments for metabolic disorders, such as obesity, type-2 diabetes, and muscle-wasting disorders associated with chronic diseases [32]. During hibernation, changes in the bear’s gut microbiota are linked to reduced production of toxins like trimethylamine N-oxide (TMAO). As reported, free-ranging brown bears exhibit elevated choline and betaine levels during hibernation, offering protective effects. This may indicate that a metabolic switch is turned on during hibernation [92]. Mastering such a switch may be of major benefit for protecting the human burden of lifestyle diseases.

Hibernators accumulate fat reserves during the summer, serving as their primary energy source during winter hibernation. Consequently, the gut microbiota composition shifts from carbohydrate-related to lipid-related, with bacteria like Pseudomonas adapting to low temperatures and secreting lipase, aiding in fat metabolism during winter fasting [34,93]. The potential impact of environmental temperature on the gut microbiota underscores concerns about climate change’s adverse effects on intestinal microorganisms. Climate change, marked by increased greenhouse gas concentrations and reduced natural carbon sinks, leads to higher temperatures. Elevated CO_2_ emissions affect soil microbial composition and function, potentially influencing the human gut microbiome through alterations in soil biodiversity that impact food crop quality [94]. In bats, it was recently reported that hibernation energy requirements could change as an adaptation to a warmer climate [95].

## 7. Impact of the SARS-CoV-2 Pandemia on Gut Microbiota

While the impact of the SARS-CoV-2 virus on the gut microbiome is still being studied, emerging evidence suggests that the virus can alter the composition and function of the gut microbiota, with potential implications for both gastrointestinal and systemic health. The SARS-CoV-2 pandemic has prompted investigations into the susceptibility of various animal hosts and the potential risk of zoonotic transmissions to humans. While bats are known reservoir hosts for SARS-CoV-2 and other viruses [96], recent studies show that domestic pigs and chickens resist intranasal or ocular/oronasal SARS-CoV-2 infection [97]. The reasons behind the absence of susceptibility of chickens and pigs to SARS-CoV-2 remain unclear. Several factors, including the absence of compatible SARS-CoV-2 receptor binding sites, may contribute to the resistance observed in pigs and chickens. Additionally, dietary factors and gut microbiota could play a role. For instance, incorporating fermented grains into pig diets enhances gastrointestinal health, regulates intestinal pH, modulates the composition of gut bacteria, and improves overall pig performance [98]. Similarly, in broiler chickens, fermented feed enhances nutrient absorption, suppresses the growth of harmful gut bacteria, and reduces anti-nutritional factors in plant proteins, leading to improved performance and gut health [99]. Considering the critical role of impaired interferon (IFN) response in severe COVID-19 cases [100], it is noteworthy that fermented dairy products from camels increase IFN-γ mRNA expression in the intestines of mice [101]. Moreover, fermented *B. rapa* extracts promoted the production of IFN-γ and interleukin (IL)-10 in mouse spleen cells more than non-fermented vegetable extracts. The salutogenic properties of these extracts appear to be mediated by Lactobacillus. Thus, as recent findings suggest a potential link between the beneficial effects of fermented foods, probiotics, and prebiotics on gut immunity and protection against severe COVID-19 infection, further investigations are warranted to make our gut microbiome better prepared for the next pandemic. 

## 8. Conclusions

Insights from the animal kingdom underscore the vital role of the gut microbiota in mitigating the impact of lifestyle diseases. Hibernating animals, for instance, display remarkable resilience and adaptability to external conditions, with their gut microbiota potentially playing a significant role in these adaptations. In certain circumstances, environmental factors influence the selection of microbial communities, leading to the development of a highly adapted and beneficial microbiome. The rapid restructuring of the microbiome can introduce new metabolic traits, aiding the organism’s adaptation and acclimatization to environmental shifts [102,103,104]. Understanding the functioning of the gut microbiota in animals and its potential for promoting health benefits provides valuable insights into modulating the gut microbiota in patients with chronic burden of lifestyle diseases to mitigate complications. 

## Figures and Tables

**Figure 1 nutrients-16-01789-f001:**
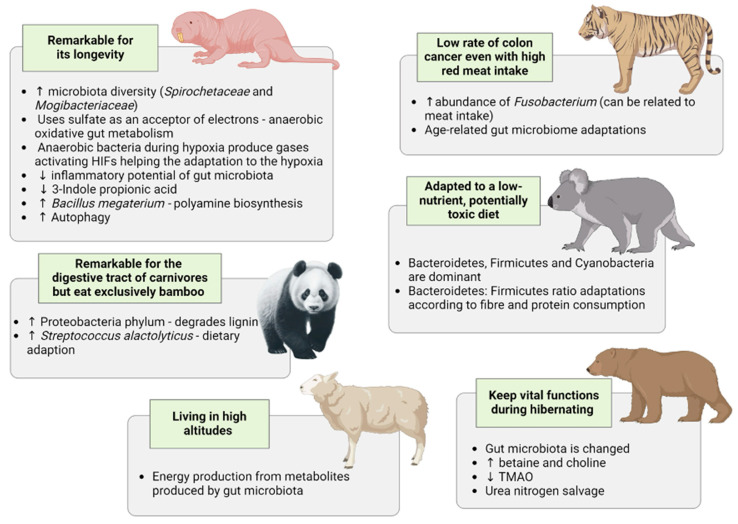
Gut microbiota and animal phenotype. Symbols: ↑ = increase;↓ = Reduction.

**Table 1 nutrients-16-01789-t001:** Examples of microbiota metabolism in animals.

References	Animals	Findings
Sommer et al. (2016) [32]	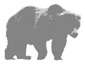 Bear	Fecal transplantation of the summer bear microbiota to germ-free mice ↑ fat gain and glucose tolerance compared microbiota from the winter.
Debebe et al. (2017) [33]	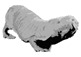 Naked Mole rat	Gut microbiome composition ↑ capacity to produce SCFA, mono-disaccharide enrichment of SCFA, and carbohydrate degradation products.
Xiao et al. (2019) [34]	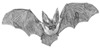 Bat	↑ Proteobacteria when compared to humans.Gut microbiotas change during early summer and late summer: Firmicutes ↓ Bacteroidetes ↑.
Zhang et al. (2018) [35]Deng et al. (2023) [36]	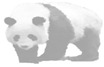 Panda	Large and diverse gene for hemicellulose hydrolysis. ↑ Proteobacteria. ↑ *Streptococcus alactolyticus*, affecting essential amino-acid biosynthesis.↑ Primary to secondary bile acid conversion by gut microbiota in old pandas.
Ilmberger et al. (2014) [37]Klinhom et al. (2023) [38]	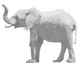 Elephant	Infant elephants: ↑ *Bifidobacterium* and *Akkermansia.*Subadult and adult elephants: ↑ Firmicutes, Bacteroidetes, and Actinobacteria.Firmicutes → lignocellulose-degrading enzymes.
Ning et al. (2020) [39]Jiang et al. (2020) [40]	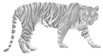 Tiger	↑ Firmicutes and *Proteobacteria*.*Collinsella*, dominant genus. *Fusobacterium* → related to meat intake.Captive and wild, different bacterial community.
Schmidt et al. (2018) [41]	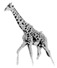 Giraffe	Giraffes from a zoo: Firmicutes and Bacteroidetes were the most abundant phyla, and Firmicutes, Bacteroides, and Spirochaetes represented the main phylum levels.
Amato et al. (2015) [30]	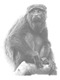 Mexicanblack howler monkeys	↑ *Ruminococcaceae* abundance in periods with low energy intake, producing more energy. ↑ genus *Butyricicoccus* with high unripe fruit intake.
Sun et al. (2016) [42]	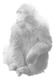 Tibetan Macaques	↑ Genus *Succinivibrio* (degradation of cellulose and hemicellulose) in the winter.↑ Genus *Prevotella* (degradation of carbohydrates) in the spring.
Wu et al. (2020) [43]	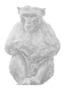 Rhesus	In high altitudes presented ↑ alpha diversity and a high Firmicutes/Bacteroidetes ratio.↑ Abundance *Christensenellaceae*, *Ruminococcaceae* (Firmicutes).
Zhang et al. (2016) [44]	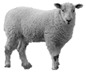 Sheep	In high-altitude ↑ SCFA producers, ↑ methane production.
Li et al. (2018) [45]Wang et al. (2020) [46]	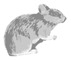 Plateau Pika	In high-altitude ↑ SCFA producers, ↑ *Prevotella* species ↑ alpha diversity. ↑ Cellulase activity in the summer.
Gibson et al. (2019) [47]	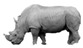 Rhino	Proteobacteria dominate the phyla in wild rhinos, followed by Bacteroidetes.
Tang et al. (2020) [48]	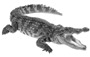 Crocodile	Firmicutes and Proteobacteria are the dominant phyla.
Song et al. (2021) [49]	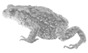 Asiatic Toad	Proteobacteria is the dominant phyla during hibernation (especially *Pseudomonas).*
Moeller and Sanders (2020) [25]	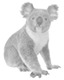 Koala	Bacteroidetes, Firmicutes, and Cyanobacteria are the dominant phyla.Koala and gut microbiota are co-adapted to a low-nutrient potentially toxic diet.

Symbols: ↑ = increase; → = stable function.

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
