# Peer review of "What Can the Gut Microbiota of Animals Teach Us about the Relationship between Nutrition and Burden of Lifestyle Diseases?"

_nutrients, 2024, doi:10.3390/nu16111789_

Round 1
Reviewer 1 Report
Comments and Suggestions for Authors
The manuscript "What Can the Gut Microbiota of Animals Teach about Nutrition and the Burden of Lifestyle Diseases?", as a non-systematic narrative review, offers a compelling perspective on advancing knowledge by upholding ancestral and evolutionary observations on GM adaptations to environmental conditions and dietary habits in animals. It is well-written and its message is both impactful and forward-thinking. I would like to suggest that the authors include a brief paragraph discussing the topic of water and its xenobiotic load as an important factor of diet while exploring its crosstalk with gut microbiota.
Author Response
Response: Thank you for your comment. We have added information about water and its xenobiotics. (Line 86-92)

Reviewer 2 Report
Comments and Suggestions for Authors
The review article submitted by Mafra et al. is an excellent piece of work, focusing on an increasingly exciting subject in nutritional terms, which is the correlation between types of nutrition and intestinal microbial composition. The authors present the work in a very coherent and particularly well-written manner. They explain the variations in the microbiome depending on dietary habits, but not only that, they also focus on the effects of industrialization itself.
The vision presented by the authors of the need to learn about what happens to the microbiome of a very well-defined group of animals, in various dietary conditions and/or in captivity versus in the wild, with a view to possible applications in improving human health, is also captivating for the reader and reflects a very different and innovative approach.
Minor comment:
Page 10 lines 380-381: the sentence "..., offering protective effects this may indicate a that a ..." needs correction.
Author Response
Response: Thank you for your comment. We have corrected it.
